# Evaluation of Scholarship Motivators and Barriers for Non-Tenure-Track Faculty in a Department of Pharmacy Practice

**DOI:** 10.3390/pharmacy11010031

**Published:** 2023-02-08

**Authors:** Cecilia Farias-Ruiz, Theresa Byrd, Eric J. MacLaughlin, Ronald G. Hall

**Affiliations:** 1Julia Jones Matthews School of Population and Public Health, Texas Tech University Health Sciences Center, Lubbock, TX 79430, USA; 2Department of Pharmacy Practice, Jerry H. Hodge School of Pharmacy, Texas Tech University Health Sciences Center, Amarillo, TX 79430, USA; 3Department of Pharmacy Practice, Jerry H. Hodge School of Pharmacy, Texas Tech University Health Sciences Center, Dallas, TX 79430, USA

**Keywords:** faculty, scholarship, research, time, pharmacy, pharmacy practice, non-tenure-track, clinical, critical thinking

## Abstract

The Accreditation Council for Pharmacy Education (ACPE) Standards 2016 set explicit expectations for faculty scholarship. However, many non-tenure-track faculty have struggled with the scholarship portion of the academic tripart mission of clinical practice, teaching, and scholarship. Therefore, we sought to identify themes regarding the barriers, motivators, and potential solutions associated with non-tenure-track faculty scholarship. Four focus group interviews were held via videoconference during July 2021, which consisted of non-tenure-track faculty within the TTUHSC Jerry H. Hodge School of Pharmacy. Each focus group answered a standard script of questions that were evaluated for face validity over a 30–60 min session. Twenty-two non-tenure-track faculty members (47% response) participated in one of the four focus group interviews. The four common barriers were insufficient time, lack of acknowledgment, obscurity of scholarship expectations, and a lack of resources and support. Scholarship’s lasting impact on academia, students, and clinical practice was the one common motivator identified by the groups. The barriers identified were not unique to our faculty, despite the unique four-city structure of our program. Actions have continued to be taken to help address the barriers and potential solutions identified by the focus groups. In summary, our results echo that non-tenure-track faculty need more time and training to help them feel like they can meet institutional scholarship requirements.

## 1. Introduction

The 2016 version of the Accreditation Council for Pharmacy Education (ACPE) Standards and Key Elements for the Professional Program in Pharmacy Leading to the Doctor of Pharmacy Degree, otherwise known as “Standards 2016”, lays out explicit expectations for faculty scholarship [1]. Meeting ACPE standards is an important function of pharmacy schools as the standards are considered the minimum criteria that need to be met to be an accredited school of pharmacy in the United States. Standards 18 and 19 are the sections of the document that outline the faculty factors evaluated by the ACPE. All of the statements within the standards regarding scholarship are outlined in Table 1.

The 2021–2022 American Association of Colleges of Pharmacy Profile of Pharmacy Faculty found that non-tenure-track faculty now represent 53% of all faculty positions and 46% at schools that offer tenure track positions [2]. Pharmacists who serve as non-tenure-track faculty members at Colleges of Pharmacy have multiple responsibilities, including didactic and experiential teaching, clinical practice, and service obligations to their school, university, and professional organizations. These faculty members are commonly referred to as “clinical faculty”. These various responsibilities often result in non-tenure-track faculty members having a relatively low distribution of effort and training in scholarship, making scholarly productivity even more difficult [3,4,5,6,7,8,9,10,11,12]. In a survey of clinical faculty members of the American College of Clinical Pharmacy Practice and Research Networks, 70% of respondents indicated they did not believe they had enough time to fulfill their nonclinical activities [13]. This has also been reported by others focused on the topic of faculty burnout [14]. Figure 1, based on data from the AACP faculty profile, demonstrates the failure to produce full professors on the non-tenure track, particularly at institutions that offer tenure track positions [2]. This discrepancy likely represents the convergence of faculty burnout and a lack of scholarship/research compared to tenure track colleagues. Others have shown that participating in writing groups may improve the confidence of non-tenure-track faculty [15]. A multimodal approach has also been suggested to help non-tenure-track faculty prioritize the important but not urgent areas of research and scholarship to advance their careers, which resulted in higher promotion rates at one institution [16].

One of the potential issues that highlights the importance of developing scholarship in non-tenure-track faculty is the high reliance on scholarship in the promotion process. One survey evaluating the assessment of pharmacy practice faculty by promotion committees found that scholarship was the primary area assessed [17]. The survey also noted that the respondents did not report significant differences in the importance of scholarship for non-tenure-track faculty. Service activities, which included clinical practice activities, were viewed as the least important factor in evaluating promotion by the respondents.

The Jerry H. Hodge School of Pharmacy is one of six schools within the Texas Tech University Health Sciences Center (TTUHSC). The TTUHSC received the prestigious Carnegie Classification for Special Focus Four-Year Research Institutions in 2022. The School of Pharmacy was established by the Texas legislature in 1993, and the first Doctor of Pharmacy class graduated in 2000. The first four-year campus of the school was built in Amarillo, with campuses for third- and fourth-year Doctor of Pharmacy students in Dallas and Lubbock. The Abilene four-year campus for the school opened in 2007. The current Dallas campus consists of a building on the Dallas Veterans Affairs Medical Center grounds, which opened in 2002, and a building in the Dallas Medical District, which was opened in 2008. The Dallas medical district locations are the primary sites for first- and second-year Doctor of Pharmacy instruction on the Dallas campus. The first four-year class that began in Dallas graduated in 2022.

The school comprises three departments: Biotechnology and Immunotherapeutics, Pharmaceutical Sciences, and Pharmacy Practice. The Department of Biotechnology and Immunotherapeutics as well as the Department of Pharmaceutical Sciences are primarily composed of tenure track faculty and have a limited number of teaching faculty who are non-tenure track. Most faculty in these departments have a PhD. A few may have a clinical doctorate instead of or in addition to a PhD. Regardless, all of the faculty in these two departments have extensive research training. On the other hand, the Pharmacy Practice Department is primarily composed of non-tenure track clinicians. These individuals have extensive practice training, but have limited formal training in regard to research [3,4,5,6]. They are primarily responsible for clinically based didactic lectures, small group facilitation, clinical laboratories (including simulation labs), and experiential instruction. The typical distribution of work among the faculty, prior to personalized discussions with their department chair, is outlined in Table 2. Non-tenure-track faculty typically have 36 weeks of clerkship instruction per year (6 rotations that are 6-weeks-long each) as part of their teaching requirements, compared to 0–12 weeks for tenure track faculty. Some non-tenure-track faculty also have clinical practice contracts, complicating their ability to engage in scholarly activities when they do not have trainees on their clinical service.

Faculty promotions at the TTUHSC Jerry H. Hodge School of Pharmacy are guided by the department-specific guidelines outlined in Table 3.

Therefore, we sought to identify themes regarding the barriers, motivators, and potential solutions associated with non-tenure-track faculty scholarship. We developed and conducted a qualitative study with TTUHSC Jerry H. Hodge School of Pharmacy, Department of Pharmacy Practice non-tenure-track faculty members. We conducted focus group interviews with these individuals to develop themes and a plan to help improve the scholarly activity of these faculty members. This work was designed to give our department qualitative feedback on what areas are working well to promote scholarship (including internal motivators) and what areas need improvement compared to other previous works that have focused on analyzing this issue from a quantitative context [4,7,13].

## 2. Materials and Methods

### 2.1. Study Overview

Four focus group interviews were held during July 2021 and involved non-tenure-track faculty of the TTUHSC Jerry H. Hodge School of Pharmacy. These were facilitated by a graduate student at the TTUHSC Julia Jones Matthews School of Population and Public Health. Department members were notified of the opportunity to volunteer as a member of one of the focus groups by email. Interested individuals notified department staff of their availability so the best dates and times to schedule the focus group interviews could be determined. Multiple focus groups were needed due to the numerous responsibilities of our non-tenure-track faculty members, including practice commitments, committee and other HSC/school services, summer didactic academic offerings, and vacations during the summer semester. Each faculty member only participated in one focus group. To encourage a comfortable environment and exchange of honest feedback, the facilitator was a non-faculty member. The sessions were conducted via videoconferencing technology (Zoom video communications), lasting 30–60 min. The sessions were recorded for transcription and analysis, with consent from each faculty member. At the beginning of each focus group session, the faculty members introduced themselves and their roles as faculty members. The introductions were conducted before recording the sessions to keep faculty member names anonymous and encourage an open conversation. During this time, the facilitator also reminded potential participants that the focus group interviews were a voluntary activity and that anyone who wished to leave the session was welcome to do so. The faculty provided informed consent orally and were advised that the sessions were audiotaped. The facilitator was under the supervision of the Chair of the TTUHSC Julia Jones Matthews Department of Public Health.

### 2.2. Data Collection

A Department of Pharmacy Practice faculty member designed open-ended questions to identify the motivators and barriers within the non-tenure-track faculty (Table 4).

The Chair of Public Health also evaluated these questions for face validity. Follow-up questions were included in the focus group session guide to assist the moderator in the clarification of questions and obtaining more detailed feedback when sufficient details were not provided naturally. Additional time was provided at the end of each focus group session to allow for other comments or suggestions that focus group members did not previously offer.

### 2.3. Coding

The audiotapes were processed through a cloud-based video platform (Kaltura, Inc.), providing the final transcriptions of each session. The initial transcriptions, which were 85–90% accurate to begin with, were then reviewed and corrected in Kaltura by the facilitator to ensure 100% accuracy within each transcription. Once the transcriptions were corrected, they were uploaded into a text file and copied into an MS Word (Microsoft, Inc.) file. Data analysis was carried out using constant comparison analysis, for which three stages of analysis were used (open coding, grouping codes into categories, and development of themes) [18]. The facilitator reviewed the transcriptions manually to complete open coding, grouped the codes into categories, and then identified general themes. The student’s supervisor reviewed the themes and the synthesis of the information based on the transcripts and discussed the findings with the student to ensure inter-rater reliability. After identifying common themes, the facilitator classified the essential themes, opportunities, and considerations provided by the faculty.

## 3. Results

Four focus groups, which consisted of 22 non-tenure-track faculty, were conducted. The groups ranged in size from 2 to 10 participants per group. The 22 participants represented 47% of the eligible non-tenure-track faculty. All of the participants were practicing pharmacists who had completed pharmacy residency training. Baseline characteristics of the focus group participants are shown in Table 5.

Analysis of the focus group text was sorted by codes and grouped by focus group number, which yielded four main barriers and one motivator (Table 6).

There was little to no engagement with questions 1, “What role do you want scholarship to play in your job”, and 2, “Should any adjustments be considered for promotion to Associate Professor? Full professor?”.

### 3.1. Barriers

#### 3.1.1. Insufficient Time

*Focus Group #1*. Emphasis was placed on the practice and teaching expectations for non-tenure-track faculty being so large that the result is insufficient time for research and scholarship. Faculty highlighted how exhausting it is to balance the myriad of expectations from the various roles expected of faculty on the non-tenure track. This sense of being overwhelmed undermines any desire to carve out time for scholarship. Scholarship requires mental freedom and being fully present to digest information and work creatively. Faculty are working to find time to deal with the exhaustion of other work areas (e.g., practice, teaching, service) for their well-being. Taking additional time to cope has resulted in less time for other activities in faculty members’ daily life: “For example, if my practice was not so overwhelming, like mentally exhausting, it is very possible for me to finish my practice requirements, go home or go to my office and still carve out maybe one or two hours to be able to do some scholarship”.

*Focus Group #2.* The balance between the time spent fulfilling the expectations of being a faculty member and that spent on small tasks that are a part of their academic teaching role (e.g., grading, writing, and entering quizzes into testing software, etc.) was discussed. It was stated that “little things add up”. Multitasking to accomplish the various roles and responsibilities associated with being a faculty member has created an imbalance between work and life. The desire to have a work–life balance was emphasized throughout the group. The group felt there was little time left for scholarship due to the various roles they are expected to fulfill. In turn, faculty purposely choose to use their precious remaining time for family time and their personal life.

*Focus Group #3.* The group discussed how they felt that there is no time set aside for scholarship. The group’s consensus was that creating time for scholarship must be a conscious decision. The issue arises when faculty are told or perceive that more pressing matters require immediate attention during their block of “free time” that was intended to be devoted to scholarship. Their discussion of the multiple responsibilities that consume most of the faculty’s time was similar to the themes focus group #2 discussed. Faculty members choose to prioritize maintaining a personal life with their remaining time. The value of work–life balance was showcased in both focus groups #2 and #3. This is a pivotal reason for the lack of scholarship amongst the non-tenure-track faculty. Lastly, focus group members mentioned faculty attrition as a stressor that negatively affects workload. Their time for scholarship has decreased as their workload in other areas (e.g., teaching, service) has increased. They felt that non-tenure-track faculty are only making meaningful progress in scholarship during the blocks they have off from experiential education rotations (e.g., two six-week blocks per year). Some faculty also felt they had more time for scholarship during the initial phase of the COVID-19 pandemic when they were not physically permitted to visit their practice site, lessening their clinical and experiential education responsibilities. Overall, they felt the model of trying to fit scholarship into limited periods of time has not been successful. They also felt that the administration has resisted changes in the curricular or practice paradigm to encourage scholarship for non-tenure-track faculty. This lack of commitment to testing shared solutions has resulted in the emotional disengagement of non-tenure-track faculty from scholarship and only “going through the motions to check boxes”.

*Focus Group #4.* There was a general agreement that the time needed to teach students and residents leaves little to no time for scholarship. These teaching responsibilities include ensuring they are prepared and successful upon graduation. Many non-tenure-track faculty have practice requirements associated with experiential education and practice contracts. The practice contracts help the department fund non-tenure-track positions but may hinder scholarly progress. This group indicated a relationship between comfort and time. They acknowledged these challenges of scholarship, which have led to a sense of procrastination and discouragement to begin or complete scholarly projects.

#### 3.1.2. Lack of Acknowledgment

*Focus Group #1*. The faculty honed in on the diversity of the faculty and how it differentiates the TTUHSC Pharmacy School from others. The participants mentioned diversity helps differentiate the school from other programs in the nation. The pharmacy school was compared to the school of medicine, where faculty are recognized or known for their expertise and publications. It is desired that the School of Pharmacy faculty be recognized for their variety of areas of expertise and contributions to their respective fields of study as well. It is also crucial that there is support offered to new faculty for them to feel equipped to engage in scholarship and balance the various time constraints of their first faculty position.

*Focus Group #2.* The expectations for scholarship and promotion were understood to be based on the number of publications a non-tenured faculty member has. However, the faculty of this group expressed a desire to expand the mold of the scholarship and promotion due to IRB complications, for example, the research carried out in [practice site redacted], “A lot of what I do is publish case reports and an abstract form, and that is kinda the standard for [practice site retracted]. We have an international meeting every year where we publish abstracts, present posters. And they’re relatively meaningful because these are things that we can’t do trials for. And I feel like it’s frowned upon”. Discouragement was felt throughout the group due to consistent scholarship efforts going unrecognized because they are different types of scholarship that do not fit the “standard mold”.

*Focus Group #3*. Scholarship requires much time and dedication, and recognition of this effort is vital in motivating faculty to pursue scholarship. Recognition may not come from within the institution but through publications. It is frustrating waiting to see whether journals will accept the work submitted. Faculty are often discouraged when their work is rejected because there is no credit for the time and dedication put into submitting it.

*Focus Group #4.* Along with focus group #2, this focus group agreed that scholarship should not be confined to a one-size-fits-all approach: “It is frustrating when they produce scholarship that is in their strengths and passion that does not fit the mold created by administrators”. The lack of recognition of scholarship is disheartening for faculty.

#### 3.1.3. Obscurity of Scholarship Expectations

*Focus Group #1.* There was a consensus amongst the faculty that the scholarship expectations were reasonable. The lack of clarity about what kind of scholarship is wanted, such as a book chapter or a peer-reviewed article, concerned faculty. It was felt institutions should communicate the preferred type of scholarship. Expectations regarding the preferred type of scholarship are vague and dependent on the personal preference of the administrator reviewing their dossiers: “So, if there were some sort of like transparency as far as like what you should be shooting for to make promotional requirements, that would be better”.

*Focus Group #2.* Some faculty felt that the evaluation of scholarship was too restrictive. Specifically, focusing on peer-reviewed articles over posters and other scholarly work is frustrating when some faculty are undergoing annual reviews or contemplating applying for promotion. The faculty in this focus group honed in on the importance of feeling comfortable and working in areas of their strengths to produce scholarship. Collaborations that consider faculty strengths can foster excitement and ease the pressure to produce scholarship. The faculty in this group were worried that their colleagues’ regional and/or national reputation would not considered in the narrow definition of scholarship as peer-reviewed publications and that, therefore, they would not meet the requirements for promotion. There was also frustration and confusion about other universities utilizing different standards for scholarship in the promotion process.

*Focus Group #3*. “Out-of-the-box innovation creates liberty in the type of scholarship faculty need to be producing to be promoted”. It was mentioned that innovation is a core value of the institution and should be rewarded. While liberty and innovation are desired, the group also expressed the need for more specific promotion criteria to guide how things are rated.

*Focus Group #4.* “The challenge that arises is the inconsistency within an institution”. It was agreed that significant frustration lies in the inconsistency of scholarship expectations across different departments within the school. This group provided the example of people with similar roles across various departments within the school not having the same number of publications required, causing frustration among those required to do more scholarship. This group expressed frustration with the variation in the definition of what constitutes scholarship and the variety of personal interpretations of what meets the guidelines for scholarship. The issue addressed were not related to the guidelines themselves but the room left for individual interpretation of what meets the guidelines.

#### 3.1.4. Lack of Resource Support

*Focus Group #1.* This group focused on insufficient resources available to non-tenure-track faculty relative to the number of non-tenure-track faculty in the department. The faculty said they feel hesitant in asking for help given that only two people in the department have Master’s-level clinical research training. This creates an actual or perceived bottleneck, often leading to long periods of waiting before help can be acquired.

*Focus Group #2.* Similar to focus group #1, there was mention of the available resources, but difficulties in utilizing them were raised. Faculty noted the example of working with the HSC-supported Clinical Research Institute, which is headquartered in Lubbock: “For example, reaching out to the Clinical Research Institute in Lubbock for help may become difficult because of the turnover in the positions and the workload, and the response time is delayed, so you wonder if that person you’re reaching out to will still be within that department by the time you get through with a manuscript”. Publication fees were also mentioned as an available resource, but the faculty voiced concerns about using the department’s money for this purpose.

*Focus Group #3.* The HSC-supported Clinical Research Institute was also mentioned in this focus group. Similarly to focus group #2, faculty faced difficulties in using this resource. The problem with long response times when reaching out for assistance due to high turnover was also noted by this group. It was mentioned that, in one case, so many people had occupied one statistician staff position in such a short amount of time that the status of an ongoing paper was lost, and currently, they are unsure where their manuscript is in the process.

*Focus Group #4*. The HSC-supported Clinical Research Institute was also mentioned in this focus group. Similar to focus group #3, the focus group concluded that the turnover time was lengthy and hurt the project’s outcome in the long run. Additionally, there was mention of the faculty’s work not being prioritized within the Clinical Research Institute. The faculty members also noted that their discouragement led them to no longer engage with the Clinical Research Institute for support.

### 3.2. Motivator

#### The Lasting Impact Scholarship Has on Academia, Students, and Clinical Practice

Scholarship creates an opportunity for innovation and allows faculty to leave a lasting impact on pharmacy practice and their students. It is a pathway to making a difference. For faculty, the freedom to innovate and be known as the first to do something was the most exciting aspect of scholarship. The opportunity scholarship provides in turning faculty’s strengths and passions into innovations that can change health care, and the opportunity to create or do something no one has done before, was a motivator. The most common word used amongst all four groups was “difference”, as in making a difference in someone’s life or pharmacy practice. Having an impact on students plays a pivotal role in the “why”, for faculty: “it may be the phone call from a student or a resident five years later. Just to say thank you for doing what you did and pushing me as a student”. Lastly, the faculty expressed that to excite people, they must be met halfway, and their efforts need to be recognized as scholarship. The faculty desired for their strengths to be considered and for more types of scholarship to be recognized for promotion purposes. They felt this would increase their drive to continue scholarship production.

### 3.3. Opportunities

#### 3.3.1. Leadership

Leadership plays a vital role in the faculty’s motivation for producing scholarship. The focus groups concluded that while the institution’s resources may play a role in the lack of scholarship production, the support they have received from departmental leadership has aided in their production of scholarship. The Vice-Chair of Research was mentioned particularly in focus group #4. His aid in providing one-on-one, individualized, hands-on meetings assisted in technical writing, wherein faculty were the least comfortable. Leadership teams who share similar interests with the faculty were also mentioned to support faculty in formulating ideas, writing protocols, and reviewing papers. The department’s Vice-Chair of Research also assists faculty with any support needed and in finding resources.

#### 3.3.2. Innovation

Below are several innovations and topics that faculty feel the HSC, or the School of Pharmacy, should be known for:Laura Bush Institute for Women’s Health: the push for gender-specific health;Geriatrics division within the School of Pharmacy;Providing care to a significant rural geographical location;Current resident projects;Teaching innovation;Experiential education;Correctional health care;Different practice settings.

The focus groups mentioned numerous times the pride they felt in the institution’s innovation, honing in on several works produced by the faculty that set the institution apart from many. The innovation aspect of scholarship motivates the faculty to produce scholarship. However, the faculty also expressed a desire to expand on existing projects so that the institution would become known and recognized for innovative work contributing to the practice of pharmacy. It was stated that the faculty is producing “really cool stuff but not publishing about it” and then attending conferences to only hear about other institutions’ novel work, covering topics that the TTUHSC has been working on for over ten years, the difference being that the TTUHSC faculty did not publish their experience, so it is novel to others. The faculty are proud of their work and the large diversity of specialties at TTUHSC. It is the diversity within the faculty, such as geriatrics, rural health care, correctional health care, and poison and toxicology, that the faculty feel the institution should capitalize on because this diversity in specialties is a differentiating factor that sets the institution apart from many others. From diversity comes diverse innovation, overall positively impacting scholarship production and the institution as a whole.

### 3.4. Needs

Regarding “What institutional supports for scholarship and research are we missing?”, the faculty suggested two means of providing additional support they felt would help with the production of scholarship: statistical analysis training and technical writing seminars. It was consistently stated throughout all four focus groups that there was a need for either statistical analysis training or access to a biostatistician. The need for technical writing seminars was also mentioned because of some of the faculty’s discomfort about writing. It was conveyed that while faculty can write research articles, the struggle lies in conveying the information so that readers without background knowledge can understand it. Technical writing seminars, wherein faculty receive training on the best ways to explain their research, may help address this issue.

### 3.5. Faculty-Mentioned Solutions

The question “What would be some potential solutions to help faculty feel like they have adequate time for scholarship and research?” led to feedback on possible solutions faculty felt would assist in producing scholarship. Table 7 outlines potential solutions, with additional explanations provided below.

*Repository of ongoing scholarly projects.* A repository could be created with ongoing scholarly projects within the school of pharmacy that other faculty members are currently working on. The repository could showcase the project purpose, the contact information of the faculty leading the project, when the project is set to be finished, and the project’s current needs. This would also aid new faculty in getting acquainted with scholarship and building networks with colleagues throughout the school of pharmacy.

*Time blocks.* Faulty suggested that time blocks in their schedule dedicated to scholarship would be a great help. During this time block, faculty would be expected to work on scholarship uninterrupted. This time would help alleviate the faculty’s barrier of insufficient time in producing scholarship. There was also mention of the faculty having the additional protected time geared towards scholarship (e.g., sabbaticals).

*Collaboration via flyers or email updates.* Throughout the focus groups, there was a desire to collaborate with other faculty on ongoing or new projects. This solution aims to consider faculty’s strengths and put groups together with complementary skills. To be effective, this idea was described as follows: “You’re not going to fix the problem just by putting people in a group, and then all of a sudden they’re productive. You have to put the right people in the group. They have to be assigned to the right things”. By allowing faculty to work in areas of their strengths and collaborate, drive, innovation, and accountability are fostered. The “how” of collaboration can be answered by providing a platform where collaboration can be cultivated, such as flyers with updated projects, or emails with ongoing or proposed projects seeking assistance.

## 4. Discussion

Our results share many similarities with those of other national surveys that had been previously conducted [4,7,13,19,20]. Our focus groups revealed four barriers and one motivator regarding the pursuit of scholarly activity. It is not surprising that non-tenure-track faculty feel demotivated in this area when they feel that the definition of success is vague and that they have insufficient time and resources to accomplish their designated responsibilities, for which they often have little or no training. This may have had something to do with the lack of engagement with the question “Tell me what role scholarship and research play in your job? What role do you want it to play?” Insecurity about one’s research credentials and skills could lead to a lack of confidence regarding scholarship. It is clear that non-tenure-track faculty appreciate the value of scholarly activity, given that the one motivator identified was the lasting impact of scholarly products on academia, students, and clinical practice. Faculty focus group participants may be more inclined to engage in scholarly activity if provided additional resources they feel would be helpful, such as a repository of ongoing projects, time blocks, and invitations to collaborate on scholarly projects.

The finding of insufficient time for scholarship in our focus groups is not surprising. A national survey published in 2014 found that 70% of clinical faculty did not feel they had adequate time for their nonclinical academic needs [13]. A recent survey of pharmacy school faculty reported an average of 1.4 publications per year for non-tenure-track faculty, which is higher than the 1 publication annually reported previously [4,7,19,21]. The survey also reported that 44% of respondents spent 10% or less actual time on research [19]. However, caution should be utilized when using average numbers of publications for pharmacy practice faculty, as one report suggested that 2% of faculty accounted for 31% of the publications [21]. The “typical” percentage of time devoted to scholarship by non-tenure-track faculty in our department is 10%. However, this had not been stated in writing until recently, and some faculty thought as little as 5% of their time was available to dedicate to scholarship. Insufficient time was also reported by 57% of respondents to an AACP survey distributed among pharmacy practice faculty members published in 2009. The 2009 survey also found that only 32% of pharmacy practice faculty members thought scholarship should be required for promotion, and 40% thought that the importance of scholarship was overemphasized [7]. One could consider that the requests for an expanded definition of scholarship in our focus groups echo the view that the current evaluation and promotion system is too strict. A lack of resources or support is further emphasized when faculty feel inadequately trained. A 2010 survey of junior pharmacy practice faculty found that fewer than 15% had completed a fellowship and that less than half felt able to meet their institution’s research expectations [20]. Our faculty also noted the importance of protecting their well-being, which has been reported by others publishing about pharmacist and faculty burnout [14,22].

Others have attempted to identify ways to help improve the scholarly activity of clinical faculty [5,23]. Before these focus group interviews, and in the 18 months since, we have tried to help address some of the issues and proposed solutions identified by our focus groups (Table 8).

As mentioned by Bosso and colleagues, an attitude considering scholarship as “optional” is antithetical to academia [5]. Ignoring scholarship can lead to adverse consequences regarding career development, developing a regional, state, or national reputation, and promotion. As mentioned by Lee and colleagues, most pharmacy practice faculty come from a clinical training environment [20]. A deep sense of responsibility for patient care is embedded in the development of an independent clinical pharmacy practitioner. Therefore, it is no surprise that non-tenure-track faculty have difficulty separating themselves from patient care activities to attend to non-urgent university duties. Similarly, scheduled teaching activities take priority during the semester, which may further detract from scholarship. The same can even be said for school or professional service activities. Scholarship is the one activity where there is little follow-up at many institutions until annual performance reviews or promotion dossier reviews are needed. One could argue that the culture of scholarship being viewed as less important for non-tenure-track faculty arises from the fact that we do not give it the same level of routine accountability as other areas of being a faculty member.

It is in non-tenure-track faculty’s best interests to be involved in scholarship or research beyond just the work produced. All faculty remaining engaged in scholarship and involving their trainees is vital to honing critical thinking skills [24,25]. As noted by Persky and colleagues, while critical thinking is one of the most desired skills of a pharmacy graduate, it is unlikely to be learned within a lesson or even a single course [26]. Therefore, continuing to be involved in the research and scholarship processes alongside pharmacy trainees helps produce better pharmacists because the experience helps sharpen their critical thinking skills. One could argue that the time required to meet the overly burdensome accreditation requirements for both schools of pharmacy and residency programs works against the goal of developing critical thinking skills. In that same vein, some pharmacy schools may be well served by investing postgraduate training funds in fellowships or graduate programs to help develop the incoming faculty’s critical thinking and research skills [27,28,29].

The findings of these focus group interviews have several limitations. One is that the focus group interviews were held in July 2021, during the COVID-19 pandemic. Therefore, many faculty may have also been experiencing burnout from their clinical duties or stressors outside of work. In addition, the findings of all focus groups are the perceptions of the individuals who chose to participate. Only 47% of eligible faculty participated in the focus group interviews. In addition, we did not confirm with the participants after the analysis whether we were correct in our synthesis of the data. These findings may not apply to our department as a whole and may not apply to other pharmacy schools, especially non-state-funded, non-research-intensive programs. Similarly, not all pharmacy schools are located in multiple cities and some may not have some of the logistical challenges of our school. However, an increasing number of programs have more than one campus, and the use of videoconference technology (e.g., Zoom) during the pandemic has increased for patient care, teaching, and research opportunities.

## 5. Conclusions

Non-tenure-track faculty at TTUHSC Jerry H. Hodge School of Pharmacy revealed four barriers to and one motivator for their scholarly activity. The barriers are not unique to our faculty, despite the unique four-city structure of our program. Actions continue to be taken to help address the obstacles and potential solutions identified by the focus groups. In summary, our results echo that non-tenure-track faculty need more time and training to feel like they can meet institutional scholarship requirements.

## Figures and Tables

**Figure 1 pharmacy-11-00031-f001:**
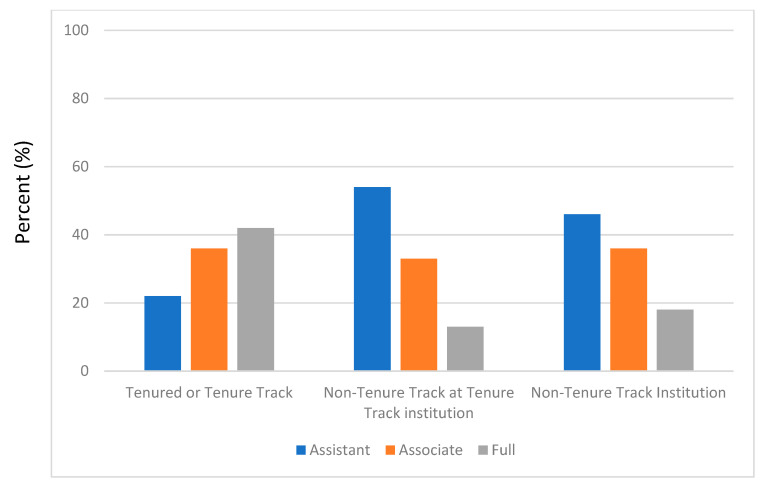
Percentage of Faculty (assistant, associate, or full professor in non-administrative roles) by Rank from the AACP 2021–2022 Profile of Pharmacy Faculty [2].

**Table 1 pharmacy-11-00031-t001:** Statements Regarding Scholarship in the 2016 ACPE Standards.

Standard	Key Element	Statement
Standard 6: College or School Vision, Mission, and Goals	6.3. Education, scholarship, service, and practice	The statements address the college’s or school’s commitment to professional education, research and scholarship, professional and community service, pharmacy practice, and continuing professional development.
Standard 8: Organization and Governance	8.2. Qualified dean	The dean is qualified to provide leadership in pharmacy professional education and practice, research and scholarship, and professional and community service.
Standard 18: Faculty and Staff—Quantitative Factors	18.1. Sufficient faculty	The college or school has a sufficient number of faculty members to effectively address the following programmatic needs: research and other scholarly activities.
Standard 19: Faculty and Staff—Qualitative Factors	19.2. Scholarly productivity	The college or school creates an environment that both requires and promotes scholarship and also develops mechanisms to assess both the quantity and quality of faculty scholarly productivity.
	19.5. Faculty/staff development	The college or school provides opportunities for career and professional development of its faculty and staff, individually and collectively, to enhance their role-related skills, scholarly productivity, and leadership.
Standard 25: Assessment Elements for Section II: Structure and Process:	25.4. Faculty productivity assessment	The college or school systematically assesses the productivity of its faculty in scholarship, teaching effectiveness, and professional and community service.

**Table 2 pharmacy-11-00031-t002:** Typical distribution of work between tenure track and non-tenure track faculty at the TTUHSC Jerry H. Hodge School of Pharmacy.

Area of Responsibility (Percent, %)	Non-Tenure Track	Tenure Track
Scholarship and Research	10	40–50
Teaching	30–35	30–35
Clinical Practice	25–50	0–10
HSC or School Service	5–10	5–10
Professional Development	5	5

Note: percentages may vary, so adding the upper limits of ranges would equal more than 100%.

**Table 3 pharmacy-11-00031-t003:** Promotion criteria used by the departments at TTUHSC Jerry H. Hodge School of Pharmacy for non-tenure-track faculty.

Department	Promotion Statement
Biotechnology and Immunotherapeutics	In general, advancement from assistant to associate and full professor will mirror the qualifications, criteria, and procedures specified for tenure track faculty in the department. However, non-tenure-track faculty members will be evaluated for promotion primarily based upon their specified job responsibility (either research or teaching) with reduced expectations in other areas. All faculty members are expected to perform academically related services as well as contribute to scholarship as part of their job duties at TTUHSC.
Pharmaceutical Sciences	Same as Biotechnology and Immunotherapeutics.
Pharmacy Practice	Promotion to Associate Professor Must demonstrate all of the following: Regional or emerging national reputation. Excellence in at least one of the assigned performance areas (teaching, scholarship, or clinical service) Proficiency in other assigned performance areas. Meaningful and measurable contribution to the professional or scientific literature. Effective and consistent service to the department and school. Promotion to Professor The faculty member must demonstrate all of the following: A scholarship record consistent with a national reputation. Consistent excellence in one of the assigned performance areas (teaching, scholarship, or clinical service). Proficiency in other assigned performance areas. Effective and consistent service to the department and school.

**Table 4 pharmacy-11-00031-t004:** Focus Group Questions.

To begin, tell me what role scholarship and research play in your job. What role do you want it to play?
2. What role does scholarship and research play in the promotion process for non-tenure-track faculty? Should any adjustments be considered for promotion to Associate Professor? Full professor? 3. Do you feel adequately trained to write review articles? Do you feel adequately trained to conduct research? If not, would you like to be?
4. What excites you about scholarship and research?
5. What are your biggest challenges to scholarship and research? What are the types of time issues you encounter when trying to work on scholarship or research? What would be some potential solutions to help faculty feel like they have adequate time for scholarship and research?
6. What institutional supportive measures for scholarship and research do you use? Are they helpful?
7. What institutional supports for scholarship and research are we missing? Which are the biggest priorities to help with your work?
8. What areas should our school be known for in the peer-reviewed literature?
9. Is there anything else you would like to tell us?

**Table 5 pharmacy-11-00031-t005:** Demographics of the focus group participants.

Characteristic	Mean (Standard Deviation)
Pharmacist (%)	100
Pharmacy Residency Training (%)	100
Female Gender (%)	73
Academic Rank (%) Assistant Professor Associate Professor Professor	64 18 18
Years at Current Academic Rank Assistant Professor Associate Professor Professor	7.5 (4.9) 5.0 (3.1) 4.0 (2.1)
Additional Research Training (%)	14

**Table 6 pharmacy-11-00031-t006:** Summary of Barriers and Motivators for Scholarly Activity.

Themes Identified by Analysis of the Focus Groups
**Barriers**
1. Insufficient time
2. Lack of acknowledgment
3. Obscurity of scholarship expectations
4. Lack of resources and support
**Motivator**
1. Scholarship’s lasting impact on academia, students, and clinical practice

**Table 7 pharmacy-11-00031-t007:** Potential solutions to providing adequate time for scholarship and research.

Repository of ongoing scholarly projects
2. Time blocks
3. Collaboration via flyers or email updates

**Table 8 pharmacy-11-00031-t008:** Measures to help improve scholarly activity.

Issue Identified	School of Pharmacy	TTUHSC
Insufficient time	Opportunity to apply to the National Institute of Health Clinical Research Scholar’s program	Opportunity to apply to the TTUHSC Clinical Research Scholar’s program
2. Lack of acknowledgment	Monthly e-mails to the department regarding faculty member publications Social media posts regarding faculty member publications Department “non-tenure-track publication of the year” awards Department-supported scholarship challenges (e.g., “Scholarship Yahtzee”)	
3. Obscurity of scholarship expectations	Two school-wide faculty development sessions related to promotion and scholarly expectations Two attempts at in-person meetings of department leadership across all four campuses to update the promotion guidelines for non-tenure-track faculty. Both attempts were canceled due to COVID-19 as the arrangements were being planned. These meetings are planning to move forward	
4. Lack of resources and support	Department-hired biostatistician dedicated to assisting with non-tenure-track faculty projects	The Clinical Research Institute has begun hiring statisticians as faculty members to decrease turnover.
5. Repository of ongoing scholarly projects	Created a Box note(online collaborative note somewhate similar to a Google Doc) for faculty members to update and interact with	
6. Time blocks	Non-tenure-track faculty continue to have two of every six clerkship blocks off each academic year	
7. Collaboration via flyers or email updates	Internal discussions have been had about the potential combinations of this suggested approach with the online repository. The department will continue exploring possible solutions to improve collaboration and communication	

## Data Availability

Data will be made available subject to TTUHSC and TTUHSC QIRB policies.

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
