# Peer review of "Evaluation of Scholarship Motivators and Barriers for Non-Tenure-Track Faculty in a Department of Pharmacy Practice"

_pharmacy, 2023, doi:10.3390/pharmacy11010031_

Round 1

Reviewer 1 Report

Dear  Editor, dear authors

Thank you for the opportunity to review your manuscript on the evaluation of scholarship motivators and barriers for non-tenure-track faculty explored through qualitative research. Several barriers, such as time, lack of recognition, misunderstood expectations and lack of support were identified. Potential measures to overcome and improve scholarly activity were presented.

As a non-tenure-track faculty member I find this research interesting, and the following comments are written in a positive manner in order to improve this work:

Introduction is short and lacking information that could serve as a comparison to other readers’ setting or perspective. For non-USA readers it may be hard to comprehend what ACPE standards are, what the difference in tenure-track and non-tenure track obligations are. Consider expanding the introduction to include more information on the setting, what the expectations of non-tenure-track faculty is (what is the ratio of clinical vs teaching obligations…), and so on… Line 40-45 explains the methods rather than the introduction.

Methods: More information is needed on participants recruitment. During coding did the facilitator consult any other researcher? Was consensus reached on all themes and topics?

Results are written in detail, and well presented

·        Were follow-up questions used for questions in line 98 for which there was no engagement?

·        How many participants were in each focus group? How were the groups formed? In  study overview you mention faculty members introduced themselves prior to the sessions. Do they not know each other, or even collaborate at the same school?

·        Participants characteristics are missing

Discussion

·        Are there any comparisons to international settings? (to make this study more appealing for international readers)

·        State reasons for not collecting participants characteristics, or if possible, collect demographics and amend the results (is there a HR liaison one could contact to gain access to such information) It is very hard to put into perspective the findings with missing characteristics which may have influenced participants answers (age, years of experience, time spent as a non-tenure-track faculty, professional/clinical background, experience in research…)

·        Line 465 and 466 are missing a reference/citation

·        Are there any plans to follow-up on participants since the introduction of measures in table 4?

·        Are suggested measures according to the ACPE?

Author Response

Please see the attached responses to your thoughtful comments.

Reviewer 2 Report

Thank you for the opportunity to review this manuscript. The role of scholarship within academia continues to be a consistent area of need and discussion. Thus, this manuscript has a relevant role.

I offer some suggestions here that can strengthen the manuscript:

1) Abstract: I recommend removing the “three legged stool” phrase and substituting with more descriptive language.

2) Introduction: While there is good information here, background and context from the literature should be considerably more in depth. Consider addressing:

*What ACPE specifically states about scholarship requirements

*True preparedness for research among residency-trained faculty

*Differences in scholarly productivity between different types of faculty (science vs practice, tenure-track vs not)

*Role of institution focus (teaching vs research-intensive) on scholarly productivity

These have all been addressed in the literature in recent years and would enhance your background information.

3) Include a gap analysis - there is no discussion of how this contributes to the literature and how it is different from other published studies. Clearly establish the gap that this is filling.

4) Why did you decide to address barriers and facilitators as an objective? How does this impact or help?

5) Materials and Methods: What is the role of research at your institution/program and in relation to faculty track type and promotion? More details on this will allow for contextualization and generalizability of the findings.

6) What training or focus group experience did the focus group facilitator have?

7) How were the focus group members recruited?

8) How was the questionnaire designed by the faculty member? Did they review the literature?

9) What was the approach for thematic analysis? Did anyone review or check the themes? Strongly consider identifying an appropriate qualitative analysis methodology and ensuring that the analysis is compliant. It may require another researcher to review or analyze.

10) Why were the faculty rank, time served at the institution, or training (fellowship vs residency) not included? That may have altered findings.

11) Consider revising the results section and condensing it. Using a table to import each of the themes and supporting quotes from each of the focus groups may make this section more focused and readable. Consider formatting consistently with other submissions in this journal.

I hope this is helpful!

Author Response

(The authors gave the same response as above.)

Reviewer 3 Report

This work is well written, well structured, but a few points may need to be clarified:

- Abstract, line 13: maybe the objectives of the work should be better stated in the abstract.

- Abstract, line 15: the verb “used” is ambiguous. Does this mean "answered"?

- Part 2.1: It is quite unclear what the focus groups are about: why 4 sessions? What is the difference between these 4 sessions? Which participants for each session, always the same? Maybe a figure illustrating the general course of this study and its various aspects would be appreciated.

- Lines 90-93: a little more information about the group would be welcome. For example, were they all pharmacists?

- Table 2: Is there not a way to give a quantitative dimension to these responses, given the number of participants in the focus groups?

- Discussion part: This was done a bit at the beginning of the Discussion section, but is it possible to find other work in the literature that addresses the issues discussed, but at the level of tenured faculty pharmacists? This is to try to get a comparison between the two categories.

Author Response

(The authors gave the same response as above.)

Reviewer 4 Report

The purpose of this study was to explore research experiences and perceptions of non tenure track (NTT) faculty at a School of Pharmacy. Overall, the work is clear and focused. The following revisions are suggested in an effort to help strengthen the work:

Title: consider changing "the" to "a"

Abstract: Make sure all results are past tense (e.g., change "are" to "were")

Introduction: This section would benefit from additional development. As a reader, I wasn't convinced that this study was needed (although i am aware this is an important issue). Why is it a problem that NTT faculty might be unproductive in scholarship? What is already known about this issue in pharm ed and other health professions (e.g., med ed, nursing ed)? What types of scholarship might these faculty conduct (e.g., clinical, education)? Etc. More text is needed to describe this issue for readers so it is clear that a) the research is needed and b) there is a gap in the literature that this study fills. As examples, here are studies/papers that might be relevant to draw from:

Fleming LW, Malinowski SS, Fleming JW, Brown MA, Davis CS, Hogan S. The impact of participation in a research/writing group on scholarly pursuits by non-tenure track clinical faculty. Currents in Pharmacy Teaching and Learning. 2017 May 1;9(3):486-90.
Glover ML, Armayor GM. An assessment of college of pharmacy promotion committees and criteria for promotion for pharmacy practice faculty. Pharmacy Education. 2006 Jul 6;6(4).
Darbishire P, Isaacs AN, Miller ML. Faculty burnout in pharmacy education. American Journal of Pharmaceutical Education. 2020 Jul 1;84(7).
Prescott WA. Facilitating advancement of clinical-track pharmacy faculty members. American Journal of Pharmaceutical Education. 2020 May 1;84(5).
Pickard AS. Towards supporting scholarship in research by clinical pharmacy faculty. Pharmacy Practice (Granada). 2006 Dec;4(4):191-4.
Behar-Horenstein LS, Beck DE, Su Y. Perceptions of pharmacy faculty need for development in educational research. Currents in Pharmacy Teaching and Learning. 2018 Jan 1;10(1):34-40.

Methods: How was the script developed? Was it inspired by any research/literature? Was it driven by concerns at the institution? Personal experiences of the research team?

Was anything done to promote trustworthiness of the findings (e.g., member-checking, auditing, peer-debriefing)? If not, this should be disclosed in the limitations of the study.

Results: Make sure all methods and results are written in past tense (except quotes, which should be presented verbatim).

Do you have any demographic information about the participants? Do they all practice pharmacy? What percent effort do they have for research? What type of research do they conduct (e.g., clinical, education, translational)? How many women participated (since gender is a known factor in research experiences)? Was the sample representative of the population at the School (beyond being NTT)? Knowing more about the sample will help readers interpret the findings and understand the extent to which the study is transferable to their own institution.

It was confusing to see the barriers results presented by Focus Group, but the other results were not. Why was this done? Consider combining the FGs in the Barriers results, unless there is something unique about each FG that warrants presenting their results separately.

Discussion: In general, this section could also be improved by situating your findings within the large body of research about NTT scholarship, etc.

As an example, pg11ln449-450: there is an entire body of literature about research training in PharmD and postgraduate trainees (and academic training programs) that could be used to support this point - and should be cited. 

Author Response

(The authors gave the same response as above.)

Round 2

Reviewer 1 Report

Dear authors

Thank you for taking your time to answer the questions and comments in detail, I believe all the additions you have made improved your manuscript. 

The only question I have is regarding the number of participant, which changed from 19 to 22. Is there a reason this happened?

Author Response

Nineteen individuals were engaged during the entire focus group. Three individuals had limited participation and therefore not included in the original number. However, upon further reflection we did not feel this was providing an accurate accounting of the number of participants. My apologies for the confusion.

Reviewer 2 Report

Thank you for carefully revising your manuscript. You have adequately answered all of the comments from the reviewers, and the manuscript has improved. Any remaining suggestions would be preferential, so I have no more comments/suggestions.

Author Response

Thank you for your time and thoughtful comments. We appreciate you noticing our efforts to address the reviewers' concerns.